# Therapeutic Strategies for Disseminated Intravascular Coagulation Associated with Aortic Aneurysm

**DOI:** 10.3390/ijms23031296

**Published:** 2022-01-24

**Authors:** Shinya Yamada, Hidesaku Asakura

**Affiliations:** Department of Hematology, Kanazawa University Hospital, Kanazawa 920-8641, Japan; hasakura@staff.kanazawa-u.ac.jp

**Keywords:** enhanced-fibrinolytic-type disseminated intravascular coagulation, serine protease, synthetic protease inhibitor, nafamostat, direct oral anticoagulant

## Abstract

Aortic aneurysms are sometimes associated with enhanced-fibrinolytic-type disseminated intravascular coagulation (DIC). In enhanced-fibrinolytic-type DIC, both coagulation and fibrinolysis are markedly activated. Typical cases show decreased platelet counts and fibrinogen levels, increased concentrations of fibrin/fibrinogen degradation products (FDP) and D-dimer, and increased FDP/D-dimer ratios. Thrombin-antithrombin complex or prothrombin fragment 1 + 2, as markers of coagulation activation, and plasmin-α_2_ plasmin inhibitor complex, a marker of fibrinolytic activation, are all markedly increased. Prolongation of prothrombin time (PT) is not so obvious, and the activated partial thromboplastin time (APTT) is rather shortened in some cases. As a result, DIC can be neither diagnosed nor excluded based on PT and APTT alone. Many of the factors involved in coagulation and fibrinolysis activation are serine proteases. Treatment of enhanced-fibrinolytic-type DIC requires consideration of how to control the function of these serine proteases. The cornerstone of DIC treatment is treatment of the underlying pathology. However, in some cases surgery is either not possible or exacerbates the DIC associated with aortic aneurysm. In such cases, pharmacotherapy becomes even more important. Unfractionated heparin, other heparins, synthetic protease inhibitors, recombinant thrombomodulin, and direct oral anticoagulants (DOACs) are agents that inhibit serine proteases, and all are effective against DIC. Inhibition of activated coagulation factors by anticoagulants is key to the treatment of DIC. Among them, DOACs can be taken orally and is useful for outpatient treatment. Combination therapy of heparin and nafamostat allows fine-adjustment of anticoagulant and antifibrinolytic effects. While warfarin is an anticoagulant, this agent is ineffective in the treatment of DIC because it inhibits the production of coagulation factors as substrates without inhibiting activated coagulation factors. In addition, monotherapy using tranexamic acid in cases of enhanced-fibrinolytic-type DIC may induce fatal thrombosis. If tranexamic acid is needed for DIC, combination with anticoagulant therapy is of critical importance.

## 1. Relationship between Aortic Aneurysm and Disseminated Intravascular Coagulation (DIC)

In 1967, Fine et al. first reported that aortic aneurysm can be complicated by DIC [1]. Over the subsequent 50 years, many reports have described cases of aortic aneurysm complicated by DIC. Although only 0.5–4.0% of aortic aneurysms show clinical symptoms such as decreased platelet count and bleeding symptoms, as many as 40% of cases show increased levels of fibrin/fibrinogen degradation products (FDP) and D-dimer [2,3]. Various hypotheses have been put forward regarding the mechanisms underlying DIC in patients with aortic aneurysms.

Some reports have revealed that radioisotope-labeled fibrinogen accumulates in the inner wall of aortic aneurysms [4] and that radioisotope-labeled platelets accumulate in the inner wall of aortic aneurysms [5]. Increased adhesion of platelets to the inner walls of aorta lacking endothelial cells or to aortic walls that have developed atherosclerosis have also been described [6,7]. Such findings may partially explain the interactions between the aortic inner wall and hemostatic factors such as fibrinogen and platelets.

In addition, the pathogenesis of DIC can be considered from a hemorheological perspective. In a normal aortic aneurysm, blood flows through the center of the aortic aneurysm, and since blood flow on the aortic wall side is slow, platelets flowing on the wall side easily adhere to the vessel wall to form a mural thrombus. Conversely, the blood in some aortic aneurysms flows in a vortex on the wall side of the aneurysm. Unlike normal aortic aneurysms, the lumen of an aortic aneurysm is stagnant with blood and is prone to the activation of coagulation. In fact, some reports have found thrombi within the incised aneurysm intraoperatively [8] or rotating within the aneurysm on ultrasonography [9]. Furthermore, on the wall side of the aorta, where blood flow is rapid, remodeling of cytoskeletal structures in the endothelial cells [10] and damage to the internal elastic lamina [11] are known to occur. Remodeling of the vessel wall by hemodynamic factors is thought to lead to changes in vessel shape, which in turn induce changes in blood flow, leading to further changes in vessel shape. In fact, a spindle-shaped aortic aneurysm was reportedly formed in a simulation in which conditions of vortex creation and morphological changes on the mural side of the aortic aneurysm were alternated [12]. In other words, rapid blood flow on the mural side within the aortic aneurysm can be considered to lead to changes in vessel shape, and these changes in vessel shape cause further changes in blood flow, leading to more stagnant blood flow in the central part of the aortic aneurysm and the gradual manifestation and worsening of coagulation activation.

Various hypotheses have sought to explain how the consumption of platelets and coagulation factors at the aneurysm site results in consumptive coagulopathy. However, since most cases of DIC associated with aortic aneurysms appear to involve enhanced-fibrinolytic-type DIC with marked fibrinolytic activation [13], those hypotheses may explain coagulation activation, but do not explain the marked fibrinolytic activation. Annexin II (annexin A2) is reportedly highly expressed in the aortic wall of a rat model of aortic aneurysm [14], while immunohistological studies showed heavy staining for annexin II in the shoulder region of atheromatous plaque from human aortic aneurysm wall tissue [15]. Annexin II is a cell surface membrane receptor that binds to both tissue plasminogen activator (t-PA) and plasminogen and dramatically enhances the activation of plasminogen by t-PA, i.e., fibrinolytic activation [16,17,18]. In other words, annexin II promotes plasmin activation by t-PA in the aortic aneurysm wall and may be involved in the pathogenesis of enhanced fibrinolysis. Annexin II is well known to be highly expressed in acute promyelocytic leukemia cells [19], but is also expressed in other cancer cells and has been associated with cancer invasion, metastasis, and angiogenesis [20,21,22]. Furthermore, macrophages within the aortic aneurysm wall reportedly show excessive production of t-PA [23,24]. Various factors are thought to be involved in the development of DIC, including coagulation activation associated with blood turbulence within the aortic aneurysm and fibrinolytic activation in the aortic aneurysm wall.

We hypothesize that coagulation activation and fibrinolytic activation in the aortic aneurysm wall does not appear after aneurysm development. Aortic aneurysms are more likely to occur in patients in whom the aorta shows innate hyperexpression of annexin II. This is a subject for future study.

## 2. Mechanism of Coagulation and Fibrinolysis

A thorough understanding of the coagulation/fibrinolysis cascade is necessary to understand the pathogenesis of DIC and to develop therapeutic strategies.

### 2.1. Coagulation Cascade 

The coagulation cascade underlies the production of fibrin clots via two quite different pathways, comprising extrinsic and intrinsic coagulation activation (Figure 1).

The extrinsic coagulation activation pathway is triggered by the expression of tissue factor from vascular endothelial cells and monocytes/macrophages following stimulation by lipopolysaccharide or cytokines, vascular injury or vascular endothelial damage. In vitro, factor VII, X, and V, prothrombin, and fibrinogen are involved in addition to tissue factor, while in vivo (as an environment where tissue factor is not as sufficient as in vitro), factor IX and are also involved. Resulting from these, thrombin and activated factors VII, IX, and X play roles as serine proteases. The coagulation cascade is shown in Figure 1. 

The intrinsic coagulation pathway begins when the blood comes into contact with collagen or a foreign body. Factors XII, XI, IX, VIII, X, V, prothrombin, and fibrinogen are involved. Resulting from these, thrombin and activated factors IX, X, XI and XII play roles as serine proteases.

Thrombin on platelet surface convert fibrinogen to fibrin. The fibrin produced by the above mechanism polymerizes to form a fibrin clot. This fibrin clot is then converted to a stabilized form by activated factor XIII (activated by thrombin). Fibrinogen level is an important factor of the structure of the fibrin network [25].

In cases of DIC associated with aortic aneurysm, the main mechanism of coagulation activation may be intrinsic coagulation activation following exposure to subendothelial collagen or foreign bodies such as stents, but this issue remains to be investigated in detail.

Similarly, whether suppression of multiple serine proteases present in the coagulation cascade or targeting and suppression of specific coagulation factors represents the best option for the treatment of DIC is an issue still to be properly investigated.

### 2.2. Fibrinolytic Cascade

After tissue repair by hemostatic thrombus, the excess thrombus is dissolved. This is achieved by t-PA (produced by the vascular endothelium) converting plasminogen (produced by the liver) into plasmin [26], which then degrades the thrombus. D-dimer represents the smallest unit of the stabilized fibrin degradation product. The degradation products of fibrinogen and pre-stabilized fibrin clots are FDPs, but not D-dimers because they do not have cross-linkages in the D fraction. Therefore, a state of marked fibrinolytic activation (as seen in enhanced-fibrinolytic-type DIC) shows an increase in the FDP/D-dimer ratio (representing a discrepancy in FDP and D-dimer levels, with a marked increase in FDP but only a mild to moderate increase in D-dimer) (Figure 2 and Figure 3). Conversely, in suppressed-fibrinolytic-type DIC, both FDP and D-dimer increase only mildly, and the FDP/D-dimer ratio does not increase (Figure 3).

Fibrinolytic inhibitors include α_2_ plasmin inhibitor (α_2_PI) and plasminogen activator inhibitor-1 (PAI-1). Whereas α_2_PI inactivates plasmin activity, PAI-1 blocks tissue plasminogen activator (Figure 2). Plasmin and α_2_PI bind in a 1:1 manner to form the plasmin_-_α_2_ plasmin inhibitor complex (PIC), which offers an indicator of the degree of fibrinolytic activation.

## 3. Classification and Laboratory Findings of DIC

DIC is a serious condition involving widespread, persistent activation of coagulation in the presence of underlying disease, resulting in diffuse microthrombi in small blood vessels [13]. Although significant activation of coagulation is always observed in DIC, the degree of fibrinolytic activation depends on the underlying disease or condition. Depending on the degree of fibrinolytic activation, DIC can be classified into three disease types (Table 1).

### 3.1. Enhanced-Fibrinolytic-Type DIC

The first of the three types is “enhanced-fibrinolytic-type DIC”, as represented by the DIC associated with aortic aneurysms. Underlying diseases causing enhanced-fibrinolytic-type DIC include hemangiomas and vascular malformations (e.g., Kasabach-Merritt syndrome [27,28], Klippel-Trenaunay-Weber syndrome [29], blue rubber bleb nevus syndrome [30,31,32]), acute promyelocytic leukemia [13,33,34,35], prostate cancer [36,37], and severe coronavirus disease 2019 (COVID-19) (Table 2) [38,39,40,41,42,43,44]. In enhanced-fibrinolytic-type DIC, multiple fibrin clots produced by marked coagulation activation dissolve one after another due to the marked activation of fibrinolysis. As a result, ischemic organ damage due to multiple microthrombi is rarely seen as a clinical manifestation [45]. In contrast, severe bleeding symptoms are more likely to occur with the dissolution of hemostatic thrombi. Characteristic laboratory findings include a low platelet count, a normal-to-prolonged prothrombin time (PT), and a shortened-to-prolonged activated partial thromboplastin time (APTT). In other words, diagnosing or excluding DIC based on PT and APTT alone is not possible. Both thrombin-antithrombin complex (TAT) (or prothrombin fragment 1 + 2 [F_1+2_]), a marker of coagulation activation, and PIC, a marker of fibrinolysis activation, are significantly increased. Due to the enhanced fibrinolysis, the FDP/D-dimer ratio is increased. In other words, FDP levels increase markedly while D-dimer levels show only a mild to moderate increase (Table 1). In addition, levels of PAI-1, a fibrinolytic inhibitor, are normal or only mildly elevated [46]. Concentrations of α_2_PI are markedly decreased, and especially when it is less than 50%, caution should be taken against major bleeding. Fibrinogen levels are also markedly decreased in typical cases, not only because of the consumption associated with the dissolution of multiple microthrombi, but also because of the degradation of fibrinogen by plasmin.

### 3.2. Suppressed-Fibrinolytic-Type DIC

The second type is “suppressed-fibrinolytic-type DIC”. This type is represented by sepsis-associated DIC, in which TAT (or F_1+2_) is markedly increased, and the plasminogen activator inhibitor-1 (PAI-I), fibrinolysis inhibitor, is also markedly increased, resulting in only mildly elevated PIC [35,47,48]. Fibrinolysis is therefore suppressed and multiple thrombi become difficult to dissolve, resulting in significant ischemic organ damage due to failure of the microcirculation. However, suppressed-fibrinolytic-type DIC is not so associated with bleeding symptoms. Fibrinogen levels are often increased, reflecting inflammation, and FDP and D-dimer levels are somewhat elevated to reflect thrombus formation, but the increase is milder than in enhanced-fibrinolytic-type DIC. Antithrombin (AT) levels are decreased, and AT supplementation should be considered, especially for levels below 70%. While levels of α_2_PI often appear normal, values are sometimes decreased (not consumptive) in the presence of hepatic insufficiency, reflecting decreased production. PT and APTT are prolonged in many cases, with more prominent prolongation of PT. The reason for this is that factor VII, which is involved in PT, has the shortest half-life among the coagulation factors. As a result, PT is likely to reflect consumption or decreased production of factor VII, and APTT is unlikely to be prolonged because of the increased production of factor VIII due to inflammation.

### 3.3. Balanced-Fibrinolytic-Type DIC

The third type is “balanced-fibrinolytic-type DIC”. This type of DIC is intermediate between the enhanced- and suppressed-fibrinolytic types. Solid tumors are a typical underlying disease for balanced-fibrinolytic-type DIC. When the balance between coagulation activation and fibrinolysis activation is suitable, hemorrhagic symptoms and organ damage are unlikely to be observed, but once the balance is broken, organ damage due to microcirculation failure or hemorrhagic symptoms become prominent. However, among the malignant tumors, prostate cancer and some other solid tumors are more often associated with enhanced-fibrinolytic-type DIC (Table 2).

## 4. Therapeutic Strategies for Enhanced-Fibrinolytic-Type DIC Associated with Aortic Aneurysm

Treatment strategies for DIC complicated by aortic aneurysm are shown in Table 3 and the flow chart in Figure 4 shows our opinion.

### 4.1. Treatment of the Aortic Aneurysm Per Se

The cornerstone of DIC treatment is the treatment of the underlying disease [49]. Prosthetic vessel replacement and stent graft endarterectomy are known treatment methods for aortic aneurysms. One perspective recommends early surgery without preoperative DIC treatment [50], while another favors surgery after achieving control of DIC [51]. Certainly, the principle of DIC treatment is to treat the underlying disease, but surgical treatment for aortic aneurysms is highly invasive and carries a high risk of mortality in some cases [52,53,54]. We believe that DIC treatment should be performed preoperatively in most cases to facilitate safer surgery.

In addition, DIC can be exacerbated by operations for the primary disease in some cases of aortic aneurysm [46,55]. The formation of a large amount of thrombus in the postoperative false lumen is thought to lead to excessive consumption of coagulation factors and exacerbation of DIC. However, postoperative exacerbation of DIC is not an indication for reoperation.

### 4.2. Follow-Up

Even if a patient with aortic aneurysm presents with DIC, not all patients are eligible for DIC treatment (Figure 4). Even in patients with obvious coagulation abnormalities, careful follow-up without medical intervention is an appropriate option if bleeding symptoms are not prominent and the risk of major bleeding in the near future appears acceptably low [46]. However, patients should be informed that they have enhanced-fibrinolytic-type DIC and are therefore prone to bleeding, and an explanation needs to be provided to their physician before any kind of invasive examination or treatment.

### 4.3. Anticoagulant Therapy

As DIC is characterized by significant coagulation activation, anticoagulation plays an important role in the treatment of DIC as well as in the treatment of the primary disease (aortic aneurysm). In clinical practice, several anticoagulants can be used for DIC, and familiarity with their mechanisms of action and side effects is indispensable.

#### 4.3.1. Unfractionated Heparin

By binding to AT, heparin dramatically enhances the inhibitory effect of AT on coagulation factor activity. AT displays broad inhibitory activity against serine proteases associated with the coagulation cascade. AT inhibits thrombin and activated factors IX, X, XI, and XII. However, in vivo, activated factor XII is mainly inhibited by C_1_-esterase inhibitor [56], and activated factor XI is mainly inhibited by α_1_ antitrypsin [57].

Unfractionated heparin can also be administered by subcutaneous injection [58,59,60] in addition to continuous intravenous infusion (1.5 to 2.5 times the control APTT) [61], and DIC treatment can be conveniently continued in an outpatient setting if the technique of home self-administered subcutaneous injection can be achieved.

However, treatment of enhanced-fibrinolytic-type DIC with unfractionated heparin alone may increase bleeding [62]. In such cases, the antifibrinolytic therapy described later may be used in combination with unfractionated heparin.

#### 4.3.2. Heparins

Heparins include low molecular weight heparins (dalteparin, enoxaparin), danaparoid, and fondaparinux. The characteristics of each agent are summarized in Table 4. All of these agents exert anticoagulant activity in an AT-dependent manner. The different heparins vary in terms of both molecular weight and half-life. The smaller the molecular weight, the weaker the AT effect and the more potent the anti-Xa activity. The higher the anti-Xa/thrombin activity ratio, the lower the risk of bleeding as a side effect, but making a general statement is difficult because these effects depend on the dosage.

(a)Low Molecular Weight Heparin (Dalteparin, Enoxaparin)

In Japan, dalteparin requires 24 h continuous intravenous infusion, whereas enoxaparin can be injected subcutaneously. However, enoxaparin is not covered by public insurance for DIC in Japan. Low molecular weight heparin is thought to have fewer bleeding side effects than unfractionated heparin, and offers the advantage of less individual variability in efficacy. In contrast, unlike unfractionated heparin, low molecular weight heparin shows the disadvantage of lacking a monitoring indicator.

The use of dalteparin [63] and enoxaparin [64,65] for enhanced-fibrinolytic-type DIC has been reported.

(b)Danaparoid

Danaparoid is usually administered intravenously twice daily, every 12 h. This agent has a long half-life of 20 h and is administered intravenously three times a week in our outpatient clinic for enhanced-fibrinolytic-type DIC. Danaparoid can be used for both acute [58,66] and chronic [67] treatment of DIC associated with aortic aneurysms. The simplicity of administration represents an advantage, but the lack of neutralizing agents and monitoring indicator are clear disadvantages.

(c)Fondaparinux

This synthetic compound comprises pentasaccharide, the smallest effective unit of heparin, and offers the advantage of requiring only once-daily subcutaneous injection. Public insurance in Japan does not cover DIC, but a report has described the use of pentasaccharide in the treatment of enhanced-fibrinolytic-type DIC associated with angiosarcoma [68]. The simplicity of administration is an advantage, but key disadvantages include the lack of neutralizing agents and monitoring indicators.

#### 4.3.3. Synthetic Protease Inhibitor

Camostat (oral), gabexate (intravenous), and nafamostat (intravenous) are known as synthetic protease inhibitors [69,70]. These agents work by binding to the substrate-binding site of serine proteases and reversibly inhibiting the interaction between the enzyme and substrate. Due to their inhibitory effects on trypsin, these agents are also used in the treatment of pancreatitis.

These synthetic protease inhibitors not only exert anticoagulant effects by inhibiting serine proteases, such as thrombin and activated factor X, but also have inhibitory effects on the serine protease plasmin [69], thus exerting antifibrinolytic effects.

Because camostat is available as an oral treatment, it has the advantage of being easily administered for DIC if it proves effective. Although some reports have described camostat as effective in DIC [71], a review of articles from the perspective of modern medicine casts doubt on its efficacy.

Gabexate has a short half-life of approximately 1 min, necessitating continuous infusion, and may be more effective in sepsis-associated DIC than in aortic aneurysm-associated DIC because it also inhibits the release of tumor necrosis factor-α (TNF-α) [72] and inhibits cytokine release from macrophages [73]. However, at clinical doses, these effects are relatively mild.

Nafamostat requires continuous infusion because of its relatively short half-life of 23.1 min [74]. In addition, the side effect of hyperkalemia [75] makes continued administration difficult in some cases. Many reports have described successful treatment with nafamostat in cases of enhanced-fibrinolytic-type DIC, including aortic aneurysms [66,76,77,78]. At clinical doses, the antifibrinolytic effect is more potent than the anticoagulant effect, and nafamostat offers a useful treatment for enhanced-fibrinolytic-type DIC in which both coagulation and fibrinolysis are markedly activated. However, the need for 24-h continuous infusion means that a change to another agent [77] is necessary if long-term treatment of DIC is required.

In addition, nafamostat has been expected to provide a useful therapeutic agent for COVID-19 for three reasons. First, nafamostat weakens the binding of severe acute respiratory syndrome coronavirus 2 (SARS-CoV-2) to the angiotensin-converting enzyme 2 (ACE2) receptor via its antiplasmin activity; when the S protein of SARS-CoV-2 is cleaved by plasmin, the ability of the virus to bind to the ACE2 receptor of the host is enhanced [79,80], Second, nafamostat inhibits SARS-CoV2 invasion into host cells by its anti-transmembrane serin protease 2 (anti-TMPRSS2) activity [79,80], as SARS-CoV-2 invasion is completed by degrading the SARS-CoV-2 S protein bound to the ACE2 receptor by TMPRSS2 [81]. Third, nafamostat show anticoagulant (AT) effects [42,44,82]. In fact, a number of clinical trials examining these issues are currently underway [83].

#### 4.3.4. Recombinant Thrombomodulin

Thrombomodulin is a high-affinity thrombin receptor that is physiologically present mainly on vascular endothelial cells. Thrombomodulin exerts anticoagulant activity by trapping thrombin. Furthermore, the thrombin-thrombomodulin complex activates protein C, which inactivates activated factor V [84,85] and activated factor VIII [86]. Recombinant thrombomodulin exerts anticoagulant effects only in the presence of thrombin, with no anticoagulant effects seen in the absence of thrombin. As a result, this agent is considered to result in fewer bleeding side effects.

Thrombomodulin also has anti-inflammatory effects [87], and recombinant thrombomodulin is highly effective in cases of DIC complicated by hematopoietic tumors and sepsis [88,89]. Because of the need for intravenous administration and the fact that long-term medication is not covered by public insurance in Japan, recombinant thrombomodulin is often used in the treatment of DIC complicated by aortic aneurysm r perioperative DIC control. Numerous reports have described the efficacy of recombinant thrombomodulin for DIC complicated by aortic aneurysms [90,91,92,93,94]. Although high DIC withdrawal rates and low risk and extent of bleeding complications are advantages, the high cost compared with other agents and the limited duration of administration available in Japan represent distinct disadvantages.

#### 4.3.5. Direct Oral Anticoagulants (DOACs)

DOACs exert direct AT-independent anticoagulant activity against activated factor X or thrombin. The characteristics of DOACs in the treatment of DIC are summarized in Table 5. However, the use of DOACs for DIC is not covered by public insurance in Japan, and caution should be exercised when using these agent.

During the choice of DOAC for DIC, dosage, patient adherence, presence of neutralizing agents, and presence of renal impairment should be considered. Each DOAC has a short blood peak (1–4 h) and half-life (half a day), so DIC has the potential to quickly worsen with a single missed dose. In patients with good adherence, twice-daily apixaban is often used to maintain stable blood levels. The insurance coverage for andexanet differs from country to country, requiring confirmation of availability. Abbreviations: DOAC, direct oral anticoagulant; DIC, disseminated intravascular coagulation.

(a)Xa Inhibitors

Among the DOACs, rivaroxaban, edoxaban, and apixaban are classified as Xa inhibitors and inhibit both free activated factor X and activated factor X bound to the prothrombinase complex in plasma.

(1)Rivaroxaban

In some cases, rivaroxaban is used from the start of DIC treatment [95,96,97], while in others, it is introduced after DIC treatment with some other anticoagulant [61,64,77].

Rivaroxaban is a very convenient treatment choice because it can be administered orally once a day to treat DIC and can be used on an outpatient basis for a long time.

(2)Edoxaban

We consider edoxaban as an easy-to-use agent for the treatment of DIC in aortic aneurysms, because dose adjustment based on age and weight is possible and aortic aneurysms are common among elderly Japanese. To date, only one case in which edoxaban was used for DIC has been reported [98].

(3)Apixaban

Apixaban is mainly metabolized by the liver and can be used safely in patients with chronic kidney disease or on dialysis [99,100]. However, the situation differs from country to country, and the Japanese package insert does not in fact allow the use of apixaban in patients with severe renal impairment. Surprisingly, when used in dialysis patients with impaired renal function, the pharmacodynamics are almost relatively close to those of healthy subjects [101]. In particular, patients with abdominal aortic aneurysms may show impaired renal function due to nephrosclerosis or enlargement of the aortic aneurysm diameter (spillover of the aortic aneurysm into the renal arteries). This agent is easy to use in patients with impaired renal function [102].

Furthermore, in trials comparing warfarin with DOACs, rivaroxaban [103], edoxaban [104], and dabigatran [105] were associated with higher frequencies of gastrointestinal bleeding. However, the frequency of gastrointestinal bleeding with apixaban was considered comparable to that with warfarin [106], and apixaban has been the agent of choice in some cases of DIC with a risk of gastrointestinal bleeding [32].
(b)Thrombin Inhibitors
(1)Dabigatran

Dabigatran is a direct thrombin inhibitor and is the only DOAC with a neutralizing agent (idarucizumab) [107]. Two reports have described the use of dabigatran in enhanced-fibrinolytic-type DIC associated with vascular malformations [65,108].

The renal excretion rate of dabigatran is as high as 80%, and care is required in patients with renal dysfunction. The disadvantage of using this agent is its low bioavailability, at 3–7%, and its efficacy varies widely among individuals [109,110].

### 4.4. Replacement Therapy

#### 4.4.1. Concentrated Platelets and Fresh Frozen Plasma

As replacement therapy for enhanced-fibrinolytic-type DIC accompanied by strong bleeding symptoms, platelet transfusion may be performed with a target platelet count of 20,000–30,000/μL or higher, and fresh frozen plasma replacement may be performed with a target fibrinogen level of 100 mg/dL or higher. However, fresh frozen plasma is often ineffective in enhanced-fibrinolytic-type DIC. This may be due to the progressive degradation of fibrinogen, in addition to consumptive coagulopathy.

The combination of anticoagulation/antifibrinolytic is much more effective than replacement therapy with fresh frozen plasma to arrest declines in blood fibrinogen (see below).

#### 4.4.2. Factor XIII Preparation

Factor XIII has the ability to form cross-linked bonds with fibrin clots and convert them into stabilized clots. Factor XIII also enhances the antifibrinolytic effect of the thrombus by incorporating into the thrombus both α_2_PI [111,112,113] and fibronectin, which contributes to wound healing [114]. Factor XIII activity is known to be decreased in DIC [90,115,116], and reports have described hemostasis achieved with factor XIII preparations in enhanced-fibrinolytic-type DIC with bleeding symptoms [116,117].

Factor XIII may be a promising option for the treatment of enhanced-fibrinolytic-type DIC under conditions where bleeding symptoms make anticoagulation difficult. We encountered one case of hemostasis after 5 days of factor XIII treatment for bleeding symptoms in a patient with thrombocytopenia due to aplastic anemia and DIC caused by aortic aneurysm [117].

### 4.5. Combination Therapy with Anticoagulation and Antifibrinolytics

When bleeding symptoms do not improve with anticoagulation alone for enhanced-fibrinolytic-type DIC, or when the fibrinolytic activation marker PIC does not decrease sufficiently (α_2_PI does not recover), anticoagulation plus antifibrinolytic therapy may be effective. However, antifibrinolytic therapy should never be administered alone (see Table 6). 

Nafamostat (24 h continuous infusion) is a relatively mild antifibrinolytic agent, and tranexamic acid (oral or 24 h continuous infusion) is a strong antifibrinolytic agent. The choice of anticoagulant or antifibrinolytic agent should be based on a comprehensive evaluation of the expected duration of DIC treatment and whether the agent should be administered intravenously or orally.

The efficacy of treatment is evaluated by the improvement of bleeding symptoms, the recovery of fibrinogen levels, decreases in the coagulation activation marker TAT (or F_1 + 2_) and the fibrinolysis activation marker PIC, decreases in FDP and D-dimer (particularly FDP), and the recovery of α_2_PI. The degree of clinical bleeding symptoms is closely related to the degree of decrease in α_2_PI. Unsurprisingly, major bleeding may occur at any time in DIC with enhanced fibrinolysis in which α_2_PI decreases to less than 50%. When tranexamic acid is administered, the metabolism of plasminogen is accelerated due to the binding of tranexamic acid to plasminogen, and plasminogen activity is reduced (often to half the previous value). If tranexamic acid is being administered orally, a decrease in plasminogen over time can be taken as confirmation that tranexamic acid is being absorbed reliably.

#### 4.5.1. DOAC and Tranexamic Acid Combination Therapy

Combination therapy with DOAC and tranexamic acid, both of which can be administered orally, is often used in patients who require long-term DIC treatment. We have experienced a case of DIC with enhanced fibrinolysis associated with vascular malformations in which combination therapy with DOAC and tranexamic acid resulted in dramatic improvement of DIC, and the patient continued treatment on an outpatient basis for more than 3 years [32].

#### 4.5.2. Danaparoid and Tranexamic Acid Combination Therapy

Combination therapy with danaparoid and tranexamic acid [118,119] has the advantage that danaparoid can be administered intravenously approximately three times a week while tranexamic acid can be provided orally. As a result, treatment can be performed in an outpatient setting. However, since the use of home heparin autologous subcutaneous injection gained coverage by public insurance in Japan in 2012, the use of danaparoid has decreased.

#### 4.5.3. Heparin and Nafamostat Combination Therapy

The combination therapy of heparin and nafamostat [44,82,120] requires hospitalization because of the continuous infusion of nafamostat. Nafamostat has anticoagulant and antifibrinolytic effects. The anticoagulant effect of nafamostat is mild, so administration of heparin compensates for the lack of anticoagulant effects from nafamostat. No reports have described use of this combination in DIC associated with aortic aneurysms, and this is a subject for further investigation.

Since some patients with severe COVID-19 present with enhanced-fibrinolytic-type DIC [38,39,40,41,42,43,44] and nafamostat also has anti-coronaviral effects [44,79,80,81], combination therapy with heparin and nafamostat appears to offer a promising strategy for the treatment of enhanced-fibrinolytic-type DIC with severe COVID-19. It is also a promising strategy for treatment.

## 5. Treatment Strategies to Avoid in Enhanced-Fibrinolytic-Type DIC Associated with Aortic Aneurysm

### 5.1. Warfarin

Warfarin inhibits the production of vitamin K-dependent proteins (coagulation factors (in ascending order of half-life): factor VII factor IX, factor X, prothrombin (II), and coagulation inhibitory factors: protein C (PC) and protein S (PS)). Specifically, by inhibiting the activities of vitamin K epoxide reductase and vitamin K quinone reductase, warfarin inhibits protein induced by vitamin K absence or antagonist (PIVKA) -VII, -IX, -X, -II, -PC, and -PS from being γ-carboxylated and converted into functionalized factors VII, IX, X, II, PC, and PS, respectively.

All the anticoagulants for DIC discussed earlier work by inhibiting the action of activated coagulation factors. Warfarin, on the other hand, decreases the coagulation factors as so-called “substrates” before they can be activated. In fulminant hepatitis, coagulation factors are almost depleted [121], and DIC develops even under such a state. In other words, in the treatment of DIC, suppression of the activity of activated coagulation factors is important, and lowering the availability of coagulation factors as substrates is not only ineffective, but also harmful by inducing fatal bleeding. In fact, one case showed exacerbation of DIC and fatal bleeding after warfarin was used in a patient with enhanced-fibrinolytic-type DIC associated with prostate cancer [122], while another showed worsened DIC during warfarin therapy for non-valvular atrial fibrillation that immediately improved when the patient was switched to rivaroxaban, a DOAC [96].

While warfarin is classified as an anticoagulant, such as other agents for DIC, but is toxic and futile for DIC. The use of warfarin as an anticoagulant for DIC is strictly contraindicated.

### 5.2. Tranexamic Acid as a Monotherapy

Tranexamic acid binds to the lysine-binding site of plasminogen, thereby inhibiting the binding of plasminogen to fibrin. This agent also acts as an antifibrinolytic by inhibiting the activation of plasminogen by plasminogen activator on fibrin. In pathological conditions where fibrinolysis is excessively activated, tranexamic acid as a single agent can be expected to have hemostatic effects. For example, tranexamic acid has been reported to improve bleeding symptoms in cases of AL amyloidosis with prolonged PT, decreased factor X activity and increased PIC [123].

In contrast, enhanced-fibrinolytic-type DIC shows not only fibrinolytic activation, but also marked coagulation activation, and single-agent administration of antifibrinolytic agents may induce fatal thrombosis [124,125]. Although success has been reported with tranexamic acid alone in enhanced-fibrinolytic-type DIC [126,127,128], this is an extremely dangerous therapy and should always be used in combination with anticoagulants.

## 6. Summary

In enhanced-fibrinolytic-type DIC associated with aortic aneurysm, activation of both coagulation and fibrinolysis is marked. Both activated coagulation factors and plasmin, which are responsible for activation of coagulation and fibrinolysis, function as serine proteases, and proper control of their functions is the basis of enhanced-fibrinolytic-type DIC. Treatment with warfarin, which suppresses the production of coagulation factors as substrates, or tranexamic acid alone, which only activates fibrinolysis, should be avoided.

## Figures and Tables

**Figure 1 ijms-23-01296-f001:**
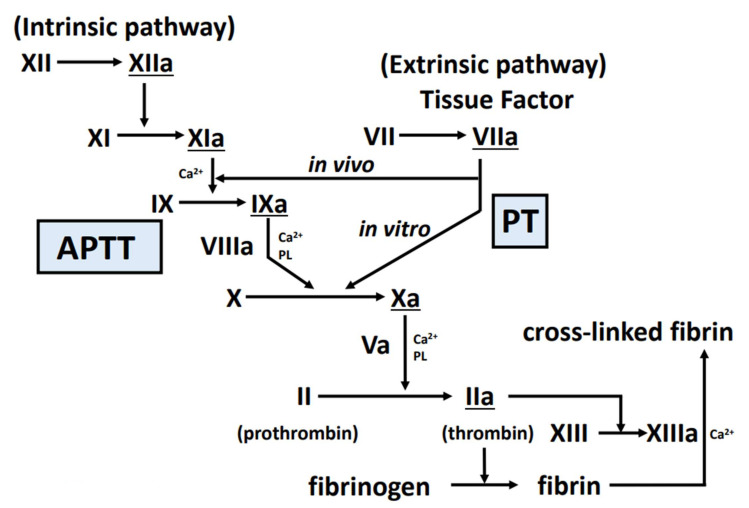
Coagulation cascade. In this figure, the cascade starting at the upper right is the extrinsic coagulation activation pathway, and the test reflecting this is PT. The cascade that begins in the upper left is the intrinsic coagulation activation pathway, and the test reflecting this is APTT. Factors XIIa, XIa, VIIa, IXa, Xa, and IIa (underlined in the figure) play roles as serine proteases. Abbreviations: APTT, activated partial thromboplastin time; PT, prothrombin time; PL, phospholipid; XIIa, activated factor XII; XIa, activated factor XI; VIIa, activated factor VII; IXa, activated factor IX; Xa, activated factor X; IIa, activated factor II.

**Figure 2 ijms-23-01296-f002:**
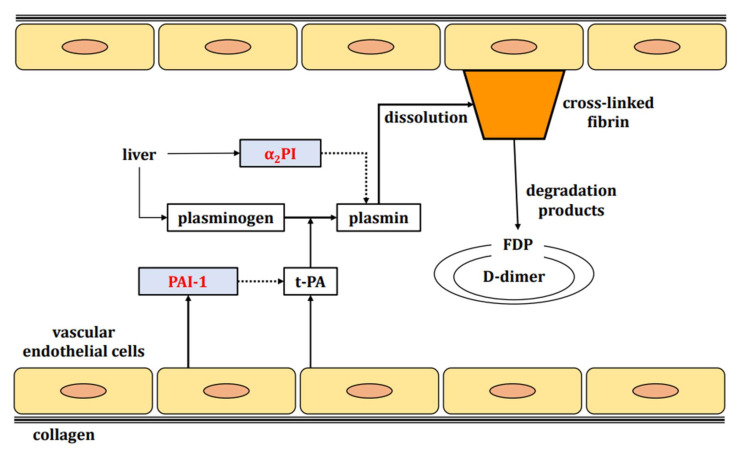
Fibrinolytic cascade. Fibrinolytic factors are indicated by black squares and fibrinolysis inhibitory factors by red letters. The liver produces plasminogen, a fibrinolytic factor, and α_2_PI, a fibrinolysis inhibitory factor. Vascular endothelium produces t-PA, a fibrinolytic factor, and PAI-1, a fibrinolysis inhibitory factor. Plasminogen is converted to plasmin by t-PA. Normally, plasmin degrades stabilized fibrin. The degradation products are FDP and D-dimer. Abbreviations: α2PI, α2plasmin inhibitor; t-PA, tissue plasminogen activator; PAI-1, plasminogen activator inhibitor-1; FDP, fibrin/fibrinogen degradation products.

**Figure 3 ijms-23-01296-f003:**
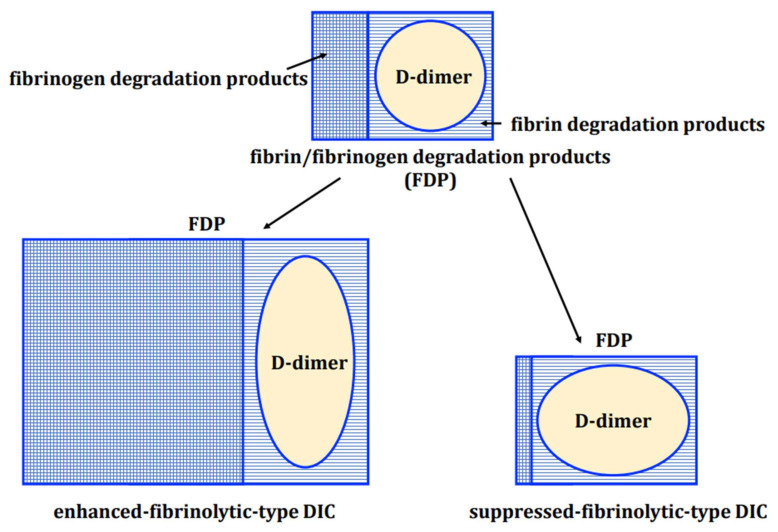
Relationship between FDP and D-dimer. FDP is a generic term overing the degradation products of both fibrin and fibrinogen. The smallest unit of fibrin degradation products (including the D-D fraction) is called D-dimer. Enhanced-fibrinolytic-type DIC shows enhanced degradation of fibrinogen, resulting in a marked increase in FDP levels. The D-dimer associated with fibrinolysis also increases, but the discrepancy between the two (reflected by an increase in the FDP/D-dimer ratio) is more pronounced. In suppressed-fibrinolytic-type DIC, fibrinogen degradation is almost absent and fibrin degradation is suppressed, so FDP and D-dimer are only mildly elevated with little deviation Abbreviations: DIC, disseminated intravascular coagulation.

**Figure 4 ijms-23-01296-f004:**
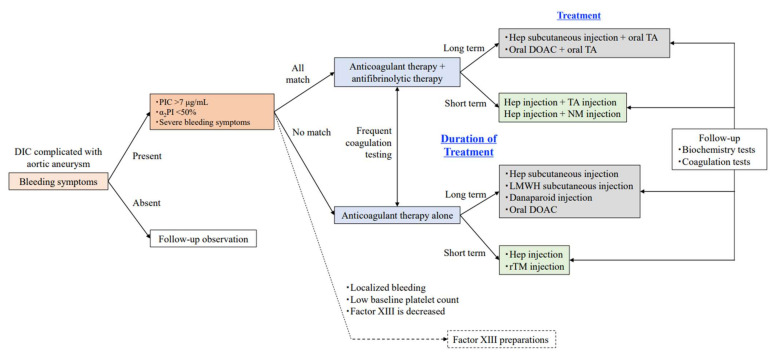
Flowchart of DIC treatment selection associated with aortic aneurysm (our opinion). Determining whether the aortic aneurysm itself can be treated is of central importance. We consider treatment strategies based on this flowchart when treating DIC associated with aortic aneurysms. We adjust the dose of the agent based on the decrease in TAT and PIC, and the recovery of fibrinogen so that the cardiovascular surgeon can safely perform the operaion. As a general rule, if no bleeding symptoms are observed (only abnormalities in coagulation tests), follow-up observation should be chosen. When bleeding symptoms are observed, if all the following conditions are met, anticoagulation and antifibrinolytics should be combined, and if none of the conditions are met, anticoagulation alone should be chosen. However, antifibrinolytic therapy should be added or the treatment regimen should be reviewed based on the results of frequent coagulation tests. Treatment should be determined according to the expected duration of therapy. If short-term treatment is planned, injectable agents with stable bioavailability and easy dose adjustment should be selected whenever possible. For long-term treatment, subcutaneous injections and oral medications should be used whenever possible. During the treatment period, blood tests and physical examinations of the whole body should be carefully performed, and patients should be followed-up frequently to check for bleeding symptoms, blood clots, and associated organ damage. If the bleeding is localized and the platelet count is low, another option is to administer a factor XIII preparation after confirming a decrease in factor XIII activity. However, since insurance coverage differs from country to country, confirmation of coverage is warranted before use. Abbreviations: DIC, disseminated intravascular coagulation; PIC, plasmin-_α2_ plasmin inhibitor complex; α2PI, α2plasmin inhibitor; Hep, unfractionated heparin; TA, tranexamic acid; NM, nafamostat mesilate; DOAC, direct oral anticoagulants; rTM, recombinant thrombomodulin; LMWH, low molecular weight heparin.

**Table 1 ijms-23-01296-t001:** DIC classification and laboratory findings.

Clinical Exams	Suppressed-Fibrinolytic-Type DIC	Balanced-Fibrinolytic-Type DIC	Enhanced-Fibrinolytic-Type DIC
Typical diseases	sepsis	solid tumor	acute promyelocytic leukemia
			aortic aneurysm
			severe COVID-19
Platelets	decreased	decreased	decreased
PT	prolonged	prolonged	normal to prolonged
APTT	prolonged	prolonged	mildly shortened to prolonged
Fibrinogen	normal to increased	decreased	markedly decreased
FDP	mildly increased	increased	markedly increased
D-dimer	mildly increased	increased	increased
FDP/D-dimer ratio	approximately 1	approximately 1–2	approximately 2–5
Antithrombin	decreased	decreased to normal	normal
TAT or F_1+2_	markedly increased	markedly increased	markedly increased
PIC	mildly increased	increased	markedly increased
α_2_ PI	normal	mildly decreased	markedly decreased
Plasminogen	decreased	mildly decreased	decreased
PAI-1	markedly increased	mildly increased	normal to mildly increased

Important findings for differentiating between disease types are highlighted in yellow. Abbreviations: DIC, disseminated intravascular coagulation; COVID-19, coronavirus disease 2019; PT, prothrombin time; APTT, activated partial thromboplastin time; FDP, fibrin/fibrinogen degradation products; TAT, thrombin-antithrombin complex; F_1 + 2_, prothrombin fragment 1 + 2; Fbg, fibrinogen; PIC, plasmin-α_2_ plasmin inhibitor complex; α2PI, α2plasmin inhibitor; PAI-1, plasminogen activator inhibitor-1.

**Table 2 ijms-23-01296-t002:** Diseases underlying enhanced-fibrinolytic-type DIC.

**1. Anomalies of the Great Vessels**
Aortic aneurysm
Aortic dissection
**2. Vascular malformation**
Kasabach-Merritt syndrome
Klippel-Trenaunay-Weber syndrome
Blue rubber bleb nevus syndrome, etc.
**3. Hematological malignancies**
Acute promyelocytic leukemia (APL)
Acute myelocytic leukemia other than APL
Acute lymphoblastic leukemia
Part of non-Hodgkin lymphoma, etc.
**4. Non-hematological malignancies**
Prostate cancer
Part of gastric adenocarcinoma
Part of colon cancer
Solid tumors with bone metastases
Malignant melanoma
Vascular-related sarcoma, etc.
**5. Early phase of heat stroke**
**6. Early phase of severe trauma**
**7. Part of severe COVID-19**

Abbreviations: COVID-19, coronavirus disease 2019.

**Table 3 ijms-23-01296-t003:** Treatment options for DIC associated with aortic aneurysms.

**1. Treatment of the Underlying Disease**
**2. Follow-up**
**3. Anticoagulant therapy**
(a) Unfractionated heparin
(b) Heparins (dalteparin, enoxaparin, danaparoid, fondaparinux)
(c) Synthetic protease inhibitors (camostat, gabexate, nafamostat)
(d) Recombinant thrombomodulin
(e) Direct oral anticoagulants (DOACs)
**4. Replacement therapy**
(a) Platelet concentrates
(b) Fresh frozen plasma(c) Fibrinogen concentrate
(d) Factor XIII preparation
**5. Anticoagulant therapy + antifibrinolytic therapy**
(a) DOACs + tranexamic acid
(b) Unfractionated heparin or Heparins + tranexamic acid
(c) Unfractionated heparin + nafamostat

Abbreviations: DIC, disseminated intravascular coagulation.

**Table 4 ijms-23-01296-t004:** Characteristics of heparins.

Agent		Unfractionated Heparin	Low Molecular Weight Heparin *		Danaparoid	Fondaparinux
			Dalteparin	Enoxaparin		
Route of Administration		iv, sc	iv	sc	iv	sc
Anti-Xa/anti-thrombin ratio		1:1	2–5:1		22:1	7400:1
Half-life (in physical chemistry)		0.5–1 h	2–4 h		20 h	17 h
Neutralizer		Protamine	Protamine (not sufficiently effective)			None
Monitoring indicator	Effect	FDP, D-dimer, TAT, F_1+2_, etc.				
	Side effect	APTT				

* Insurance coverage and dosages of low molecular weight heparin differ from country to country, necessitating confirmation of both coverage and dosage before use. Abbreviations: iv, intravenous injection; sc, subcutaneous injection; FDP, fibrin/fibrinogen degradation products; TAT, thrombin-antithrombin complex; F_1+2_, prothrombin fragment 1 + 2; APTT, activated partial thromboplastin time.

**Table 5 ijms-23-01296-t005:** Characteristics of DOACs.

Mechanism of Action	Direct Factor Xa Inhibitor			Direct Thrombin Inhibitor
Agent	rivaroxaban	edoxaban	apixaban	dabigatran
Rules of use	once a day	once a day	twice a day	twice a day
Peak blood level	2–4 h	1–2 h	1–4 h	1–3 h
Half-life (in physical chemistry)	8–11 h	5–11 h	8–15 h	12 h
Renal excretion rate	35%	50%	26%	80%
Monitoring	Not established			
Neutralizer	andexanet			idarucizumab

During the choice of DOAC for DIC, dosage, patient adherence, presence of neutralizing agents, and presence of renal impairment should be considered. Each DOAC has a short blood peak (1–4 h) and half-life (half a day), so DIC has the potential to quickly worsen with a single missed dose. In patients with good adherence, twice-daily apixaban is often used to maintain stable blood levels. The insurance coverage for andexanet differs from country to country, requiring confirmation of availability. Abbreviations: DOAC, direct oral anticoagulant; DIC, disseminated intravascular coagulation.

**Table 6 ijms-23-01296-t006:** Characteristics of anticoagulation and antifibrinolytics combination therapy.

Agent		Antifibrinolytic	
		Tranexamic Acid	Nafamostat
Anticoagulant	UFH or heparins *	Easy to adjust the balance of anticoagulation and antifibrinolysis. Treatment at home is possible if a subcutaneous UFH or heparins are selected	Easy to adjust the balance of anticoagulation and antifibrinolysis
	DOACs	Oral treatment is possible. Suitable for patients needing long-term treatment	Nafamostat is used to add anticoagulant and antifibrinolytic effects to a single DOAC

* Heparins: dalteparin, enoxaparin, danaparoid, fondaparinux. Combinations of heparin(s), DOACs, tranexamic acid, and nafamostat for combination anticoagulation and antifibrinolysis are characterized. In the acute phase, the combination of heparins and either tranexamic acid (injection) or nafamostat is dose-adjustable and useful for balancing anticoagulation with antifibrinolysis. When long-term combination therapy is required, the combination of DOACs and tranexamic acid (oral), both of which can be administered orally, or the combination of unfractionated heparin and tranexamic acid, if self-subcutaneous injection is possible, is easy to implement. Abbreviations: UFH, unfractionated heparin; DOAC, direct oral anticoagulant.

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
