# Peer review of "Therapeutic Strategies for Disseminated Intravascular Coagulation Associated with Aortic Aneurysm"

_ijms, 2022, doi:10.3390/ijms23031296_

Round 1

Reviewer 1 Report

The authors describe a very interesting review of therapeutic managements for disseminated intravascular coagulation associated with aortic aneurysm. DIC is a life-threatening complication of several disease states including sepsis, cancer, obstetrical complications, trauma and aortic aneurysm.

The manuscript is well structured, but some parts are missing some important facts that authors should add.

Page 3, lines 112-114

The authors should more accurately characterize the final stage of hemostasis. Activation of thrombin on the platelet surface leads to the conversion of fibrinogen to fibrin [25]. The conversion of fibrinogen to fibrin is the last step in the blood clotting cascade. An important fact is that the level of fibrinogen is a major determinant of the structure of the resulting fibrin network. References are missing in this part of the manuscript. This important statement was published in a new evaluation, which the authors should cite: ,, Diagnostics 2021, 11(11), 2140; https://doi.org/10.3390/diagnostics11112140“

Page 4, lines 122-124

Authors should state, that release of tissue plasminogen activator (t-PA) from endothelial cells leads to the conversion of proenzyme plasminogen into plasmin. This statement was published in a manuscript that the authors should cite: ,, Biomedicines 2020, 8(12), 605; https://doi.org/10.3390/biomedicines8120605“

Page 13, lines 428-436

The authors do not mention fibrinogen concentrate, which is used in massive bleeding DIC. They should make this statement and at the same time cite the manuscript in which it was described: ,, J Intensive Care. 2014 Feb 20;2(1):15. doi: 10.1186/2052-0492-2-15.“

Figures and tables in the text are very clearly written.

I have to say that with these 126 references of which more references are older than 5 years, it is advisable to add newer references.

Author Response

・In title, we corrected a misspelling from “intravascularcoagulation” to “intravascular coagulation”.

・In line 470, we revised “these agent” to “these agents”.

・I mistake the order of the reference number as follows:

     ×58→〇61, ×59→〇58, ×60→〇59, ×61→〇60

Response to Reviewer 1 Comments

Point 1: Page 3, lines 112-114

The authors should more accurately characterize the final stage of hemostasis. Activation of thrombin on the platelet surface leads to the conversion of fibrinogen to fibrin [25]. The conversion of fibrinogen to fibrin is the last step in the blood clotting cascade. An important fact is that the level of fibrinogen is a major determinant of the structure of the resulting fibrin network. References are missing in this part of the manuscript. This important statement was published in a new evaluation, which the authors should cite: ,, Diagnostics 2021, 11(11), 2140; https://doi.org/10.3390/diagnostics11112140“

Response 1: Thank you for your comment. I revised line 114 “Thrombin on platelet surface convert fibrinogen to fibrin. The fibrin produced by the above mechanism polymerizes to form a fibrin clot. This fibrin clot is then converted to a stabilized form by activated factor XIII (activated by thrombin). Fibrinogen level is an important factor of the structure of the fibrin network [26].”

Point 2: Page 4, lines 122-124

Authors should state, that release of tissue plasminogen activator (t-PA) from endothelial cells leads to the conversion of proenzyme plasminogen into plasmin. This statement was published in a manuscript that the authors should cite: ,, Biomedicines 2020, 8(12), 605;https://doi.org/10.3390/biomedicines8120605“

Response 2: Thank you for your comment. I add reference 26 in line 127

Point 3: Page 13, lines 428-436

The authors do not mention fibrinogen concentrate, which is used in massive bleeding DIC. They should make this statement and at the same time cite the manuscript in which it was described: ,, J Intensive Care. 2014 Feb 20;2(1):15. doi: 10.1186/2052-0492-2-15.“

Response 3: Thank you for your comment. I revised Table 3. I added “Fibrinogen concentrate”

Point 4:

I have to say that with these 126 references of which more references are older than 5 years, it is advisable to add newer references.

Response 4:Thank you for your comment. I added two more referenecs (reference 25 and 26).

Reviewer 2 Report

In this review entitled “Therapeutic strategies for disseminated intravascular coagulation associated with aortic aneurysm”, Yamada and Asakura describe the mechanisms of disseminated intravascular coagulation associated with aortic aneurysm and posible therapeutic strategies.

This is an extensive and very well written review. The topic is very interesting because of the lack of knowledge.

Major point

The same authors have just published a review of this topic based on the discussion of clinical cases  in Int J Haematol 2021 (doi: 10.1007/s12185-020-03028-z).  The recommendatios by the experts are similars. In fact, the authors using identical phrases and similar tables, such as table 1 and 3.

Minor points

The abstract is limited to describing the limitations in the diagnosis and treatment of this type of DIC. I miss in the abstract specific recommendations of the diagnosis and hemostatic treatment that could attract the attention of potential readers.

The authors indicate that determining whether the aortic aneurysm itself can be treated is of central importance. Clear indications for surgery and how to manage peri-surgical hemostasis would be welcome.  

Also greater specificity in the recommendations of the type of anticoagulant would be appreciated. That is, if you use an anticoagulant treatment, which one and why.

Author Response

・In title, we corrected a misspelling from “intravascularcoagulation” to “intravascular coagulation”.

・In line 470, we revised “these agent” to “these agents”.

・I mistake the order of the reference number as follows:

     ×58→〇61, ×59→〇58, ×60→〇59, ×61→〇60

Response to Reviewer 2 Comments

Point 1: The same authors have just published a review of this topic based on the discussion of clinical cases  in Int J Haematol 2021 (doi: 10.1007/s12185-020-03028-z).  The recommendatios by the experts are similars. In fact, the authors using identical phrases and similar tables, such as table 1 and 3.

Response 1: Thank you for your comment. As you pointed out, Table 1.3 have some similarities to the previous report, we did not describe PAI-1 in previous report, we described PAI-1 in Table 1. Also, we added anticoagulant therapy + antifibrinolytic therapy in Table 3.

Point 2: The abstract is limited to describing the limitations in the diagnosis and treatment of this type of DIC. I miss in the abstract specific recommendations of the diagnosis and hemostatic treatment that could attract the attention of potential readers.

Response 2: We added in line 23 “Among them, DOACs can be taken orally and is useful for outpatient treatment. Combination therapy of heparin and nafamostat allows fine-adjustment of anticoagulant and antifibrinolytic effects.”

Point 3: The authors indicate that determining whether the aortic aneurysm itself can be treated is of central importance. Clear indications for surgery and how to manage peri-surgical hemostasis would be welcome.

Response 3: Thank you for your comment. We added in line 245 “We adjust the dose of the agent based on the decrease in TAT and PIC, and the recovery of fibrinogen so that the cardiovascular surgeon can safely perform the operation.”

Point 4: Also greater specificity in the recommendations of the type of anticoagulant would be appreciated. That is, if you use an anticoagulant treatment, which one and why.

Response 4: Thank you for your comment. It is difficult to recommend a specific anticoagulant. Therefore, the treatent strategy is shown in Figure 4.

Round 2

Reviewer 1 Report

The presented manuscript has been corrected in response to the suggestions. The authors have followed the recommendations of the reviewer. After the revision, the provided data and addition of the results became more clear. I would like to thank the authors for resubmitting the manuscript and explaining the obscure points from the previous version.

Reviewer 2 Report

The authors have correctly answered the doubts raised